# Antimicrobial Efficacy of Fruit Peels Eco-Enzyme against *Enterococcus faecalis*: An In Vitro Study

**DOI:** 10.3390/ijerph17145107

**Published:** 2020-07-15

**Authors:** Hetal Ashvin Kumar Mavani, In Meei Tew, Lishen Wong, Hsu Zenn Yew, Alida Mahyuddin, Rohi Ahmad Ghazali, Edmond Ho Nang Pow

**Affiliations:** 1Department of Restorative Dentistry, Faculty of Dentistry, The National University of Malaysia, Kuala Lumpur 50300, Malaysia; hetalmav81@ukm.edu.my (H.A.K.M.); wonglishen@ukm.edu.my (L.W.); hz_yew@ukm.edu.my (H.Z.Y.); ehnpow@hku.hk (E.H.N.P.); 2Department of Family Dentistry, Faculty of Dentistry, The National University of Malaysia, Kuala Lumpur 50300, Malaysia; alida@ukm.edu.my; 3Department of CITRA & Teaching, Faculty of Health Sciences, The National University of Malaysia, Kuala Lumpur 50300, Malaysia; rohi@ukm.edu.my; 4Division of Restorative Dental Sciences, Faculty of Dentistry, University of Hong Kong, Hong Kong, China

**Keywords:** eco-enzyme, endodontics, *Enterococcus faecalis*, fruit peels, root canal irrigants, sodium hypochlorite

## Abstract

Sodium hypochlorite (NaOCl), an effective endodontic irrigant against *Enterococcus faecalis* (EF), is harmful to periapical tissues. Natural pineapple-orange eco-enzymes (M-EE) and papaya eco-enzyme (P-EE) could be potential alternatives. This study aimed to assess the antimicrobial efficacy of M-EE and P-EE at different concentrations and fermentation periods against EF, compared to 2.5% NaOCl. Fermented M-EE and P-EE (3 and 6 months) at various concentrations were mixed with EF in a 96-well plate incubated for 24 h anaerobically. Minimum inhibitory concentration (MIC) and minimum bactericidal concentration (MBC) of M-EE and P-EE were determined via EF growth observation. EF inhibition was quantitatively measured and compared between different irrigants using the one-way analysis of variance (ANOVA), and different fermentation periods using the independent-samples T-test. M-EE and P-EE showed MIC at 50% and MBC at 100% concentrations. There was no significant difference in antimicrobial effect when comparing M-EE and P-EE at 50% and 100% to 2.5% NaOCl. P-EE at 6 months fermentation exhibited higher EF inhibition compared to 3 months at concentrations of 25% (*p* = 0.017) and 0.78% (*p* = 0.009). The antimicrobial properties of M-EE and P-EE, at both 100% and 50% concentrations, are comparable to 2.5% NaOCl. They could therefore be potential alternative endodontic irrigants, but further studies are required.

## 1. Introduction

Endodontic treatment aims to eradicate infection and prevent reinfection within the root canal system. However, the complete elimination of debris and bacteria in the root canal system is impossible because of the complexity of the root canal anatomy [1]. The remnant bacteria, especially *Enterococcus faecalis,* are believed to be the most resistant microorganism that contribute to the persistent periradicular lesion after root canal treatment [2]. Different virulence factors, such as aggregation substance, lipoteichoic acid, and pheromones, have facilitated *Enterococcus faecalis* to survive in dentinal tubules up to 400 μm depth in a nutrient deficiency ecology environment and cause secondary infection [3].

The use of endodontic irrigants as an adjunct to mechanical debridement during root canal treatment is often emphasized for optimizing root canal disinfection. Among all endodontic irrigants, sodium hypochlorite (NaOCl) is widely accepted as the “gold standard”, because of its potent antibacterial and proteolytic activity. Its antibacterial efficacy, especially against *Enterococcus faecalis*, has been well established [4]. It is primarily attributed to chlorine, which induces irreversible oxidation of bacterial enzyme and disrupts the metabolic function of bacterial cells [5]. Nevertheless, its toxic effects on vital tissues in the event of accidental extrusion beyond root canal space have been documented. Damage could irreversibly occur in periradicular tissues involving soft tissue spaces and the neurovascular structures [6].

Natural plant extracts have been studied as a potential substitute for NaOCl as endodontic irrigants [7,8,9,10]. Fruit peels have displayed antimicrobial activities against a wide range of microorganisms, including *Enterococcus faecalis* [11,12]. Following fermentation, the antibacterial properties of fruit peels are further enhanced as organic substances are decomposed, yielding secondary metabolites known as bioactive compounds or phytochemicals [13,14]. The extraction of enzymes, organic acids, and phenolic compounds through the fermentation process is preferred over conventional methods that require costly solvents, involve the possible degradation of heat-labile compounds, and through which it is difficult to obtain high purity extracts [15]. Thus, fermented fruit peels, known as eco-enzyme, could be an alternative endodontic irrigant.

An eco-enzyme extracted from fermented unripe papaya (*Carica papaya*) peels is found to be rich in papain, which exhibits significant antibacterial efficacy against *Enterococcus faecalis* [16]. A study by Duarte and co-workers reported 0.8% of papain is equally effective as 1.0% NaOCl in inhibiting *Enterococcus faecalis* growth [17]. It has less harmful effects on vital tissues compared to NaOCl, as its proteolytic activities selectively target unhealthy tissues where α1-antitrypsin plasmatic antiprotease is absent [18]. Besides, phytochemicals found in the papaya peel eco-enzyme demonstrate a potential anti-inflammatory effect, which minimizes the chronic inflammatory process and tissue destruction, particularly in cases of apical periodontitis [19].

Similarly, eco-enzyme derived from pineapple (*Ananas comosus*) and orange (*Citrus aurantium* L.) peels have been shown to have antimicrobial as well as anti-inflammatory properties [20]. The synergistic effect of the two eco-enzymes increases the potency of their antimicrobial activity against a wide range of bacteria [21]. The high content of polyphenolic compounds and flavonoids in pineapple and orange peel extracts are found to be responsible for their excellent antimicrobial and antioxidant activities [22,23]. Bromelain from pineapple extracts is shown to be effective in killing *Enterococcus faecalis* by disrupting the peptidoglycan and polysaccharide components of bacterial cell membranes [24].

Numerous previous studies have looked into the antibacterial properties of different endodontic irrigants available on the market [25,26,27,28]. However, studies on eco-enzyme fermented from fruits and/or its peel as alternative endodontic irrigants are lacking. Hence, this study aimed to investigate the potential antibacterial activity using fermented fruit peel wastes (a mixture of pineapple-orange peel extracts and papaya peel extracts) at different concentrations and fermentation periods against *Enterococcus faecalis* in comparison to NaOCl.

## 2. Materials and Methods

Ethical approval was obtained from the research ethical committee of The National University of Malaysia (UKM PPI/111/8/JEP-2018-660) to conduct this in vitro study.

### 2.1. Preparation of Eco-Enzyme Extracts

Two types of eco-enzyme extracts were prepared in this study according to the method described by Arun and Sivashanmugam (2017) [20]: (1) a mixture of orange peel (*Citrus aurantium*) and pineapple peel (*Ananas comosus*) eco-enzyme extract (M-EE) at four to six ratio and (2) papaya peel (*Carica papaya*) eco-enzyme extract (P-EE). Each eco-enzyme extract was prepared in triplicates by mixing 75 g fruit peels, 25 g molasses, and 250 mL tap water in airtight containers. Each of the mixtures was stored and fermented for 3 and 6 months. M-EE and P-EE with 100% concentration after the fermentation period were sterile filtered and further diluted to a concentration of 50%, 25%, 12.5%, 6.25%, 3.13%, 1.56%, and 0.78%, respectively, for antimicrobial efficacy test against *Enterococcus faecalis*.

### 2.2. Bacterial Strain

*Enterococcus faecalis* ATCC^®^ 29212^TM^ bacterial strain was used in this study. The inoculum was prepared based on the Clinical and Laboratory Standards Institute (CLSI) protocol (2012) [29]. *Enterococcus faecalis* was cultured in brain-heart infusion (BHI) broth and incubated at 37 °C with 95% relative humidity for 24 h. The bacterial cell density was further adjusted to 0.5 McFarland Standards (1–2 × 10^6^ CFU/mL).

### 2.3. Determination of Minimal Inhibitory Concentration (MIC)

Minimum inhibitory concentration (MIC) of both M-EE and P-EE extracts of different concentrations against *Enterococcus faecalis* was determined using CLSI (2012) protocol [29]. In brief, 50 μL of each of the following solutions: 2.5% NaOCl (positive control), BHI broth (negative control), M-EE and P-EE with different concentrations at a 3 and 6 months fermentation period, were mixed with 50 μL *Enterococcus faecalis* in a 96-well plate (Biologix, Selangor, Malaysia) and incubated under the anaerobic condition for 24 h at 37 °C. All samples were prepared in triplicates. The bacterial growth was observed with naked eyes and the appearance of the well plate was recorded as “no turbidity observed” (no bacterial growth) or “turbidity observed” (bacterial growth). Bacterial growth was also quantitatively measured using ELISA microplate reader (Thermo Fisher Scientific, Waltham, MA, USA) at 625 nm wavelength. Mean optical density (OD) of positive, negative, and tested groups were recorded and compared with its corresponding untreated control wells.

### 2.4. Determination of Minimal Bactericidal Concentration (MBC)

Minimal bactericidal concentration (MBC) of M-EE and P-EE against *Enterococcus faecalis* was determined by taking samples with no apparent bacterial growth in wells from MIC tests and cultured on BHI agar plates. After an incubation period of 24 h at 37 °C with 95% relative humidity, the agar plates containing M-EE and P-EE were inspected with the naked eye for any bacterial colonies growth.

### 2.5. Statistical Analysis

Results were analyzed with the data collected from M-EE and P-EE of different concentrations at a 3 and 6 months fermentation period, NaOCl, and BHI broth, using Statistical Package for the Social Sciences (SPSS) Version 23 software (IBM, Armonk, NY, USA). The results from direct visualization in MIC and MBC tests were descriptively documented. The comparisons of mean difference in OD between M-EE and P-EE of different concentrations with 2.5% NaOCl (positive control group) and BHI broth (negative control group) at the two fermentation periods were tested using one-way analysis of variance (ANOVA). A univariate ANOVA was used to test within-subject effects and Bonferroni multiple comparisons were performed to detect differences in mean OD difference over the M-EE, P-EE, positive and negative control groups. The comparison of the mean difference between two fermentation periods for each tested group was analyzed using the independent-samples T-test. All tests were set at a significant level of 0.05.

## 3. Results

The MIC of M-EE and P-EE at various concentrations over 3 and 6 months fermentation periods are shown in Table 1 and Table 2. Both M-EE and P-EE showed *Enterococcus faecalis* growth inhibition at concentrations of 50% and 100% at a 3 and 6 months fermentation period under direct visualization (Figure 1 and Figure 2).

The mean difference in optical density (ΔOD) of 2.5% NaOCl, BHI broth, and M-EE of various concentrations at a 3 and 6 months fermentation period were compared and are illustrated in Figure 3. The ΔODs of M-EE with 50% and 100% concentration at both fermentation periods were significantly lower compared to that of the BHI group (*p* < 0.05) and no significant difference was found when compared with that of the NaOCl group (*p* > 0.05). No significant differences in ΔOD were found between the 3 months and 6 months fermentation groups across all concentrations of M-EE (*p* > 0.05).

The results of ΔOD between 2.5% NaOCl, BHI broth, and P-EE of various concentrations at a 3 and 6 months fermentation period are summarized in Figure 4. No significant difference in ΔOD was found when comparing the 50% or 100% P-EE group with the NaOCl group (*p* > 0.05). Significantly higher efficacy against *Enterococcus faecaelis* was found in the 25% (*p* = 0.017) and 0.78% (*p* = 0.009) concentrations of P-EE at 6 months fermentation group compared to the 3 months fermentation group (*p* < 0.05).

The M-EE and P-EE at 50% and 100% concentrations with no apparent *Enterococcus faecalis* growth in MIC plate wells were further tested for MBC and the results are presented in Table 3. Only M-EE and P-EE at 100% concentration had bactericidal activities against *Enterococcus faecalis*.

## 4. Discussion

Studies on the possible applications of natural substances in dentistry have increased remarkably in the last decade. Pineapple, orange, and papaya eco-enzymes have been widely investigated for therapeutic use in the management of periodontal diseases and caries removal [30,31,32,33,34]. To the best of our knowledge, this is the first in vitro study investigating the potential use of pineapple-orange and papaya eco-enzymes as endodontic irrigants.

MIC determination for M-EE and P-EE using the broth microdilution method in this study provides quantitative evaluations of their respective antimicrobial agents against *Enterococcus faecalis* [29,35], as compared to the agar disk-diffusion method in previous studies [36,37,38]. The broth microdilution method is preferred over the agar disk-diffusion method, as the former has better reproducibility and the confounding factor of diffusion potency of antimicrobial agents impregnated discs can be eliminated [39].

The concentration of NaOCl commonly used as endodontic irrigant ranges from 0.5% to 6.0% [40,41,42]. However, its optimal concentration for endodontic treatment is controversial [43]. Despite the fact that the usage of 5.25% NaOCl is known as the gold standard [44], the choice of 2.5% NaOCl as a positive control in this study is based on its equivalent antimicrobial efficacy to that of 5.25% NaOCl [45]. This is supported by another study that demonstrated no significant difference in *Enterococcus faecaelis* biofilm eradication on dentine surface treated with 2.5% and 5.25% NaOCl [46]. Moreover, NaOCl’s cytotoxicity is concentration dependent [47]. A NaOCl concentration of 2.5% has been shown to be less toxic to periapical tissues without compromising its antimicrobial properties [48]. Hence, the lowest possible clinically effective concentration should be used to ensure patient safety [49].

Although NaOCl appears to be the primary choice of endodontic irrigant, it cannot remove inorganic components of the smear layer which could only be removed by chelators, such as ethylenediaminetetraacetic acid (EDTA) [50]. However, when NaOCl is combined with EDTA, the availability of free chlorine in NaOCl is reduced and thus the antimicrobial efficacy of NaOCl against *Enterococcus faecalis* is compromised [51]. On the other hand, the addition of EDTA into the fruit peels eco-enzyme can enhance the enzyme proteolytic activities. This is in agreement with an in vitro study, which demonstrated the highest bromelain activity when bromelain was extracted using EDTA from pineapple peels [52]. This is also supported by another study by Chaiwut et al. [53], which investigated proteolytic components of papaya peel and reported that proteolytic activities of protease were activated by the addition of chelating agents such as EDTA.

The findings from this study showed that the antimicrobial efficacy of both M-EE and P-EE was concentration dependent. M-EE and P-EE with a minimum concentration of 50% had a comparable bacteriostatic effect on *Enterococcus faecalis* with 2.5% NaOCl, but the bactericidal effect was observed only in full strength M-EE and P-EE. These results are in agreement with previous studies that concluded better antimicrobial effects against endodontic pathogens at higher concentrations of fruit eco-enzyme [54,55]. With at least 50% concentration of M-EE or P-EE, hydrolytic enzymes, especially protease and amylase in M-EE and protease in P-EE, destroy the physical integrity of extracellular polymeric substances (EPS), the structure of *Enterococcus faecalis*, and lead to cell death [56].

Acetic acid, which was derived from the natural fermentation of fruit peels, also contributed to its antimicrobial properties. Though the concentration of acetic acid in M-EE and P-EE was not measured in this study, a previous study showed that its concentration increases with a longer fermentation period [57]. This was attributed to the hydrolysis of complex organic compounds into simpler compounds through anaerobic fermentation, resulting in the accumulation of low molecular weight acetic acid. Acetic acid can cross bacterial cell membranes because of the pH gradient, leading to the disruption of cellular metabolic activities of bacteria [58]. The higher osmotic pressure within bacterial cells also leads to water influx and cellular osmolysis [59].

A three-month fermentation period is a minimum prerequisite for fruit eco-enzyme preparation to achieve an optimal concentration of hydrolytic enzymes and acetic acid [60]. It is postulated that higher hydrolytic enzyme and acetic acid levels after a longer fermentation period would help to enhance the antimicrobial effects of fruit eco-enzyme [61]. The findings of P-EE conform to this theory as better antimicrobial efficacy was generally observed at a 6 months fermentation period compared to that of 3 months. Antimicrobial activity of papain is mainly related to enzymatic actions, such as amidase and esterase, which improves in a more acidic environment with a fermentation period of more than 3 months [62,63]. However, the aforementioned findings were not replicated in the current study of M-EE, which showed no significant enhancement of *Enterococcus faecalis* inhibition in fermentation periods longer than 3 months. This might be because the maturation of M-EE with peak hydrolytic enzyme occurred at 3 months fermentation, as a result of synergistic interactions between pineapple and orange peels eco-enzymes [64].

Ethanol, a byproduct of fruit peel fermentation, may theoretically have antibacterial efficacy against a wide range of pathological microorganisms. However, it has been shown that 25% of ethanol is required to inactivate *Enterococcus faecalis* effectively [65]. Natural fruit peel fermentation produces a low concentration of ethanol [66] of merely 5.4% to 13% at best, with the addition of dry yeast [67,68]. As this potential confounding factor is considered to be insignificant in previous studies, the alcohol concentration was hence not measured in the present study.

The pH of a solution may alter the antibacterial property of endodontic irrigants. However, the pH of our eco-enzyme was not determined in the present study. Fermented fruit peels eco-enzymes are generally acidic with pH ranges from 2.8 to 3.6 [69,70], and this pH itself does not have significant antimicrobial properties against endodontic pathogenic microorganisms [58], including *Enterococcus faecalis* [71]. This finding is also in line with a study that reported that *Enterococcus faecalis* was able to tolerate the acidic environment at pH 2.9–4.2 [72].

There are several limitations to this study. Although endodontic infection is often polymicrobial, the pathogen studied in this research was confined to *Enterococcus faecalis*. Hence, an assumption could not be made as to whether eco-enzymes would exhibit similar antimicrobial activities against other endodontic pathogens. The study could be further improved by having another control group of fruit peels extract without fermentation, for a better comparison of active compounds which may affect antimicrobial efficacy against *Enterococcus faecalis*.

Further research is needed to explore the antimicrobial properties of the tested fruits in their natural state without fermentation as well as using various parts of the fruits as their antimicrobial nature has been established. Analysis by high-performance liquid chromatography (HPLC) is suggested to identify the major compounds contributing to the antimicrobial properties of the fruits. Purifying the identified compounds with potent antimicrobial agents could then be explored pharmaceutically for commercial purposes.

## 5. Conclusions

The present study showed that the concentration of P-EE and M-EE at 50% and above exhibited significant antibacterial activity against *Enterococcus faecalis*. These results suggest that P-EE and M-EE could be exploited as a potential alternative to NaOCl as endodontic irrigants. However, further in vivo and clinical studies are required.

## Figures and Tables

**Figure 1 ijerph-17-05107-f001:**
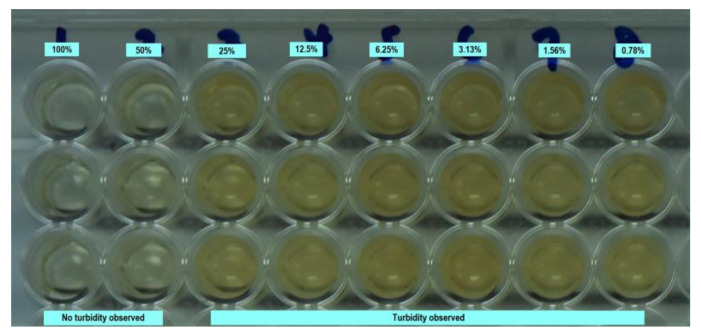
Observation of *Enterococcus faecalis* growth in a 96-well plate at different concentrations of M-EE fermented for 3 months.

**Figure 2 ijerph-17-05107-f002:**
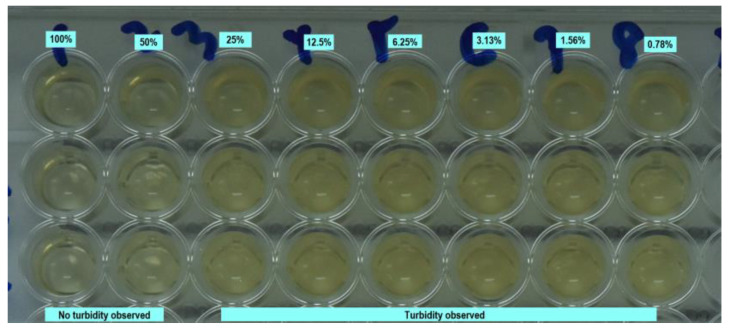
Observation of *Enterococcus faecalis* growth in a 96-well plate at different concentrations of P-EE fermented for 3 months.

**Figure 3 ijerph-17-05107-f003:**
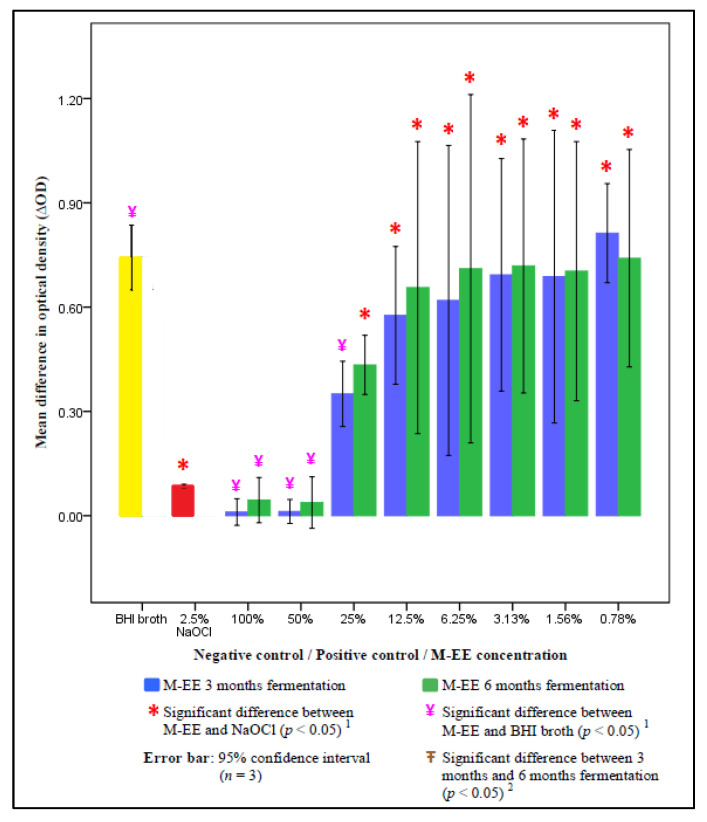
Mean difference in optical density between M-EE of different concentrations at 3 and 6 months fermentation, NaOCl, and BHI broth. ^1^ One-way analysis of variance (ANOVA): Difference between M-EE, NaOCL, and brain-heart infusion (BHI) broth (*p* < 0.05). ^2^ Independent-samples T-test: Difference between fermentation periods (*p* < 0.05).

**Figure 4 ijerph-17-05107-f004:**
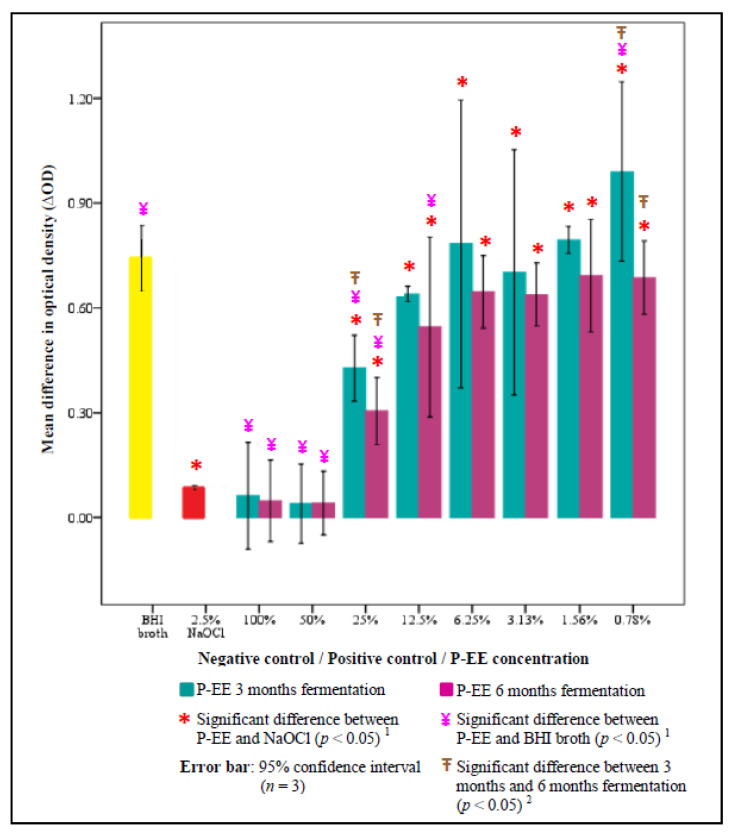
Mean difference in optical density between P-EE of different concentrations at 3 and 6 months fermentation, NaOCl, and BHI broth. ^1^ One-way ANOVA: Difference between P-EE, NaOCL, and BHI broth (*p* < 0.05). ^2^ Independent-samples T-test: Difference between fermentation periods (*p* < 0.05).

**Table 1 ijerph-17-05107-t001:** *Enterococcus faecalis* growth at different concentrations of pineapple-orange eco-enzymes (M-EE).

Concentration (%)	100	50	25	12.5	6.25	3.13	1.56	0.78
3 months fermentation	1	1	2	2	2	2	2	2
6 months fermentation	1	1	2	2	2	2	2	2

1: No turbidity observed. 2: Turbidity observed.

**Table 2 ijerph-17-05107-t002:** *Enterococcus faecalis* growth at different concentrations of papaya eco-enzyme (P-EE).

Concentration (%)	100	50	25	12.5	6.25	3.13	1.56	0.78
3 months fermentation	1	1	2	2	2	2	2	2
6 months fermentation	1	1	2	2	2	2	2	2

1: No turbidity observed. 2: Turbidity observed.

**Table 3 ijerph-17-05107-t003:** Sensitivity against *Enterococcus faecalis* of M-EE and P-EE fermented for 3 and 6 months at 50% and 100% concentrations.

Type of Endodontic Irrigant	Fermentation Period (months)	Concentration (%)	Sensitivity against *Enterococcus faecalis*
M-EE	3	100	1
		50	2
	6	100	1
		50	2
P-EE	3	100	1
		50	2
	6	100	1
		50	2

1: Sensitive. 2: Resistant.

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
