# Peer review of "Antimicrobial Efficacy of Fruit Peels Eco-Enzyme against *Enterococcus faecalis*: An In Vitro Study"

_ijerph, 2020, doi:10.3390/ijerph17145107_

Round 1
Reviewer 1 Report
The authors of this manuscript attempted to investigate in vitro study the antimicrobial efficacy of fruit peels eco-enzyme against Enterococcus faecalis.
The study was generally well written and interpretation of results. But, there are major comments regarding experimental conducted.
Major comments
1. (Introduction)
Line 67 “Numerous previous studies” you have only two, please read (Diogo P, Mota M, Fernandes C, et al. Is the chlorophyll derivative Zn(II)e6Me a good photosensitizer to be used in root canal disinfection?. Photodiagnosis Photodyn Ther. 2018;22:205-211. doi:10.1016/j.pdpdt.2018.04.009 and Basrani B, Tjäderhane L, Santos JM, et al. Efficacy of chlorhexidine- and calcium hydroxide-containing medicaments against Enterococcus faecalis in vitro. Oral Surg Oral Med Oral Pathol Oral Radiol Endod. 2003;96(5):618-624. doi:10.1016/s1079-2104(03)00166-5)
- (Introduction)
The introduction is well described; however, you need to justify why you used fermentation products! Fermentation is not covered in the introduction.
- (M&M) How was the sample calculated?
- (M&M) Fermentation for what you refer to in your discussion produces acetic acid.
what is the concentration of this acid in the different dilutions?
- (M&M) I think it was crucial to have a group without fermentation? And know the pH of the groups?
. (Abstract) How can you justify that the antimicrobial action is by the fruit peels eco-enzyme And not by the products of the fermentation of the sugar of the fruit?
Reviewer 2 Report
I read with great interest the manuscript entitled: "Antimicrobial efficacy of fruit peels eco-enzyme 3 against Enterococcus faecalis: An in vitro study" and the topic is very interesting since despite sodium hypochlorite produces a harmful effect on periapical tissues and the use of new irrigators with a natural origin and higher biocompatibility may be interesting. The study is appropriate and may be published in IJERPH.
However, some minor corrections should be performed, for example:
- Beware of formatting. Please, pay attention to punctuation marks, the appearance of capitalized words, behind commas, double space between words. Also, the English style should be checked. ("the mostly commonly," for instance, in line 44)
KEYWORDS:
- Should appear in alphabetical order
RESULTS:
- Please, add the p values to all the tables and indicate the presence of significant differences in tables
DISCUSSION:
- In my view, it is too short. It should be a little more elaborate. Discuss the use of other irrigants, possible interactions between M-EE/P-EE with chelating agents such as EDTA
REFERENCES:
Please check the format in All references. It is wrong. Follow the instructions from IJERPH: Author 1, A.B.; Author 2, C.D. Title of the article. Abbreviated Journal Name Year, Volume, page range.
Author Response
TABLE OF RESPONSE TO REVIEWER 2’ S COMMENTS
|
No |
REVIEWER’S COMMENTS |
RESPONSES TO COMMENTS |
|
1. |
Formatting -pay attention to punctuation marks, the appearance of capitalized words, behind commas, double space between words, English style. -line 44 “the mostly commonly” |
Formatting has been checked. Grammar has been checked using ‘Grammarly’. |
|
2 |
Keywords Should appear in alphabetical order |
Keywords have been arranged in alphabetical order (page 1, Line 30-31). |
|
3 |
Results Please add the p values to all the tables and indicate the presence of significant differences in tables |
Table 1, 2 and 3 represented observational findings of ‘turbid’ or a ‘non-turbid’ appearance. Statistical analysis was not applied. |
|
4 |
Discussion It should be a little more elaborate. Discuss the use of other irrigants, possible interactions between M-EE/P-EE with chelating agents such as EDTA. |
Potential use of EDTA has been added into the discussion – its function in endodontics and its interaction with eco-enzymes (page 7, line 237-246). |
|
5. |
References Please check the format in All references. |
Format has been checked and corrected. |
Round 2
Reviewer 1 Report
The manuscript assessed in vitro study about Antimicrobial efficacy of fruit peels eco-enzyme against Enterococcus faecalis. This research is under the scope of this journal; the topic is relevant for readers and this research deals with potentially significant knowledge to the field. The article was good presentation and easy to read. Aim of this paper is quite interesting. The introduction is direct to the objective that you want to study, the research is well done.
However, there are numerous issues in the present manuscript that need to be addressed before publication:
Line 145 “Figure 3. 145 ΔOD of M-EE” correct the text.
Line 169 - The legend of Table 3 “Sensitivity against Enterococcus faecalis” change for Enterococcus faecalis. And M-EE and P-EE should describe the meaning!
The reference is on final of the sentence. But, the authors et al. (Chaiwut et al. (47)) in the text of the manuscript, references should come immediately afterwards.
The references are not standardized (example ref 24 and 35- the titles are all in italics) and need to be correctly inserted in the manuscript.
Author Response
TABLE OF RESPONSE TO REVIEWER 1’ S COMMENTS (round 2)
|
REVIEWER’S COMMENTS |
RESPONSES TO COMMENTS |
|
Line 145 “Figure 3. 145 ΔOD of M-EE” correct the text.
Line 169 - The legend of Table 3 “Sensitivity against Enterococcus faecalis” change for Enterococcus faecalis. And M-EE and P-EE should describe the meaning!
The reference is on final of the sentence. But, the authors et al. (Chaiwut et al. (47)) in the text of the manuscript, references should come immediately afterwards.
The references are not standardized (example ref 24 and 35- the titles are all in italics) and need to be correctly inserted in the manuscript. |
No correction done as “Figure 3.” is the last 2 words of previous sentence. (Page 4, Line 148).
“Enterococcus faecalis” has been changed to Enterococcus faecalis (italic) (Page 6, Line 171).
Reference has been moved as advised (Page 7, Line 199).
The reference format has been corrected for Reference no 29 & 35 (Page 10, Line 346-348 & 364-367). |